# Endocrinological Involvement in Children and Adolescents Affected by COVID-19: A Narrative Review

**DOI:** 10.3390/jcm12165248

**Published:** 2023-08-11

**Authors:** Valeria Calcaterra, Veronica Maria Tagi, Raffaella De Santis, Andrea Biuso, Silvia Taranto, Enza D’Auria, Gianvincenzo Zuccotti

**Affiliations:** 1Department of Internal Medicine and Therapeutics, University of Pavia, 27100 Pavia, Italy; 2Department of Pediatrics, Vittore Buzzi Children’s Hospital, 20154 Milan, Italy; veronica.tagi@gmail.com (V.M.T.); raffaella.desantis@unimi.it (R.D.S.); andreabiuso@gmail.com (A.B.); silvia.taranto@unimi.it (S.T.); enza.dauria@unimi.it (E.D.); gianvincenzo.zuccotti@unimi.it (G.Z.); 3Department of Biomedical and Clinical Sciences, University of Milan, 20122 Milan, Italy

**Keywords:** SARS-CoV-2, COVID-19, thyroid disease, adrenal glands, hypothalamus, hypophysis, precocious puberty, gonad, diabetes, children, adolescents

## Abstract

Since the advent of the severe acute respiratory syndrome coronavirus 2 (SARS-CoV-2) pandemic, an increased incidence of several endocrinological anomalies in acute-phase and/or long-term complications has been described. The aim of this review is to provide a broad overview of the available literature regarding changes in the worldwide epidemiology of endocrinological involvement in children since December 2019 and to report the evidence supporting its association with coronavirus disease 2019 (COVID-19). Although little is known regarding the involvement of endocrine organs during COVID-19 in children, the current evidence in adults and epidemiological studies on the pediatric population suggest the presence of a causal association between the virus and endocrinopathies. Untreated transient thyroid dysfunction, sick euthyroid syndrome, nonthyroidal illness syndrome, and hypothalamic–pituitary–adrenal (HPA) axis and central precocious puberty have been observed in children in acute infection and/or during multisystem inflammatory syndrome development. Furthermore, a higher frequency of ketoacidosis at onset in children with a new diagnosis of type 1 diabetes is reported in the literature. Although the direct association between COVID-19 and endocrinological involvement has not been confirmed yet, data on the development of different endocrinopathies in children, both during acute infection and as a result of its long-term complications, have been reported. This information is of primary importance to guide the management of patients with previous or current COVID-19.

## 1. Introduction

Early after the first description of severe acute respiratory syndrome coronavirus 2 (SARS-CoV-2), there was strong evidence of a milder clinical presentation of the infection in most children, with 90% of cases being asymptomatic, mild, or moderate [1]. The prevalence of severe coronavirus disease 2019 (COVID-19) in children also remained low after the spread of new variants with greater transmissibility, like delta and omicron [2]. These findings have been attributed to several immunological mechanisms distinguishing children from adults, including a higher production of interferons at the mucosal surface, which rapidly alert the immune system at the first sign of infection [3]; a faster innate immune response due to their mostly untrained T cells, which are more likely to respond to novel viruses; and expression of acquired specific antibodies or memory cells through previous exposure to the endemic coronaviruses that commonly circulate among infants [4].

Despite their less severe manifestations in the acute phase, two main long-term complications of COVID-19 have been reported in children: multisystem inflammatory syndrome in children (MIS-C) and long COVID.

MIS-C is a serious and potentially fatal complication involving the cardiovascular system and other organs such as the stomach, liver, and intestines, with an estimated incidence of 316 every 1,000,000 children infected with SARS-CoV-2 before the availability of vaccines [5]. Since the symptoms typically begin four to six weeks after the initial infection, it has been hypothesized that the virus remains in children’s gut, causing an irritation in its mucosa. For this reason, viral antigens manage to pass through the gut’s barrier to circulation, reaching other organs and triggering a major inflammatory response [6].

Long COVID is a heterogeneous multisystemic condition characterized by the persistence of signs and symptoms that occur three months from the onset of COVID-19 and last at least two months and cannot be explained by an alternative diagnosis [7]

Lopez-Leon et al. [8] conducted a systematic review, showing a prevalence of long COVID in pediatric patients of 25.24%. The most frequently reported symptoms were mood alterations (16.50%), fatigue (9.66%), sleep disorders (8.42%), headache (7.84%), and respiratory symptoms (7.62%). Similar to adults, the detected risk factors for the development of long COVID in children are older age, female sex, severe COVID-19, overweight/obesity, comorbid allergic diseases, and other long-term comorbidities [8]. No guidelines exist to address long COVID diagnosis and management. The underlying pathogenetic mechanisms and correct management of these patients have not yet been defined [8].

SARS-CoV-2 is known to interact with the host cells through its spike protein by binding to the membrane enzyme angiotensin converting enzyme 2 (ACE2). Once the virus enters the cells, the STAT3/NF-kB pathway is activated, causing the production of proinflammatory cytokines and chemokines, which leads to a systemic hyperinflammation known as “cytokine storm” [9].

ACE2 is not expressed only by the lung cells: it is ubiquitarian, which explains the multiorgan impairment typically observed in COVID-19 [10].

Endocrine tissues express ACE2 as well, especially at the level of the thyroid, ovaries and testis, and this may be responsible for the endocrinological involvement during SARS-CoV-2 infection [11,12]. Since the advent of the pandemic, there has been increases in the incidence of several endocrinological anomalies and in the severity of their manifestations in both adults and children [13].

The aim of this narrative review is to provide an overview of the available literature regarding the epidemiology of the endocrinological involvement in children since December 2019 and to report the evidence supporting its association with COVID-19. To revise data on the endocrine disorders during SARS-CoV-2 infection, it is useful to define adequate pediatric long-term monitoring.

## 2. Materials and Methods

A narrative review on the potential connections between SARS-CoV-2 infection and endocrinological impairment in children was performed. We conducted an extensive literature search on PubMed and Embase, with language restricted to English only and publication dates between 2019 and present time (1 May 2023), divided by different organs or systems. We included in our review all types of articles (original article, review, metanalysis, case report, case series, and clinical practice guideline) regarding children aged 1–18 years with a proven ongoing or previous SARS-CoV-2 infection or diagnosis of MIS-C who experienced an impairment of one or more of the main endocrine organs or systems involved in children with COVID19: thyroid, adrenal glands, hypothalamus–pituitary system, gonads and pancreas. Articles focusing only on adults were excluded from our review. The list of keywords used for each analyzed endocrine system or organ and the respective type and number of articles found are summarized in Table 1. Starting from a total of 424 papers screened by title/abstract, the authors reviewed the full texts of relevant articles (*n* = 190).

## 3. Thyroid

Very limited studies and reports describe the association between SARS-CoV-2 infection and thyroid disease in pediatric subjects [14,15]. McCowan et al. [14] conducted a study on 244 children with anomalies in thyroid function, either hypothyroidism or hyperthyroidism, in a tertiary pediatric endocrine center in the United Kingdom before and after COVID-19 to identify any change in their presentation. Despite an unchanged rate of thyroid dysfunction before and after the pandemic, they observed an increase in the number of cases of untreated transient thyroid dysfunction. They speculated that this finding may be due to the development of thyroiditis secondary to SARS-CoV-2 infection, which regressed before requiring treatment [14].

A retrospective chart analysis on 233 children screened for TSH level in New York did not find any significant difference in the frequency of TSH abnormalities prior and after the pandemic started. However, it is unknown who had a prior COVID-19 infection or who received anti-SARS-CoV-2 vaccination among them [16].

A cross-sectional analytical study assessed the levels of free thyroxine (fT4), fT3, TSH and inflammatory markers, especially interleukin-6 (IL-6), in hospitalized children and adults with moderate-to-severe COVID-19 infection [17]. In total, 67.7% of all patients and 78% of the pediatric population had an abnormal thyroid profile, with sick euthyroid syndrome, which is characterized by low fT3 levels with normal TSH levels, associated with a systemic illness, being the most frequent. This status was associated with a significantly higher risk of death and severe inflammation (detected using high IL-6 levels), highlighting the importance of monitoring thyroid function test results in patients with severe COVID-19 infection [17]. The higher reported incidence of thyroid dysfunction in comparison to other studies, where most analyzed patients presented with mild COVID-19, suggests a correlation of the severity of the disease with thyroid gland impairment [15,18]. In line with these data, a retrospective study on pediatric patients referred to our center and diagnosed with MIS-C revealed that more than 90% of patients exhibited nonthyroidal illness syndrome (NTIS), defined as any abnormality in thyroid function (TF) tests (FT3, FT4, and TSH) in the presence of critical illness and absence of a pre-existing hormonal abnormality [19]. Among the NTIS laboratory profile variants, an isolated decrease in the level of fT3 was the most common [19]. Concordant data were obtained from a subsequent prospective cohort study on a population of 43 children and adolescents with MIS-C, where 79% of patients presented low FT3 levels, which, in 10 cases, was associated with abnormal values of fT4 and/or TSH. Twenty months later, a restoration of TH balance was observed in 100% of patients [20].

The association of MIS-C with thyroid disfunction was also studied in another cohort of 46 children and an aged-matched control group. Also in this case, NTIS was reported in 97.2% of MIS-C cases. Moreover, the FT3 levels were lower in patients with MIS-C who were admitted to the ICU with worse clinical presentations [21].

Although the available data regarding the involvement of thyroid function during COVID-19 infection in childhood are still limited, the current evidence supports the theory that the virus plays a certain role in thyroid dysfunction, which is transitory in most cases. Therefore, given the heterogeneity of dysthyroid manifestations, it would be useful to routinely assess thyroid function indices in patients with a COVID-19 infection, especially if they present with moderate to severe disease signs and MIS-C in order to start, when needed, the appropriate treatment as soon as possible [13].

It has been hypothesized that SARS-CoV-2 may enter the thyroid cells via ACE2 and transmembrane serine protease (2TMPRSS2), which is highly expressed in this gland [22,23]. It is well known that viral-infections-related thyroiditis causes preformed colloid release and consequently impairs raised thyroid hormone concentrations. Furthermore, COVID-19 may destroy follicular cells through an autoimmune mechanism [24]. Indeed, virus-induced cytokine storm is characterized by hyperactivity of the Th1/Th17 immune response with overexpression of proinflammatory cytokines, such as IL-6, which has been demonstrated to be strongly associated with thyroiditis [25]. The latter has been tested in adult populations, assessing patients’ thyroid antibody status. A study conducted in India revealed positivity of anti-TPO antibodies in 13.6% of the participants [26]. A similar frequency was observed in a large group of European patients, 23.6% of whom were seropositive for at least one thyroid autoantibody [27]. Regarding this hypothesis, Flokas et al. [28] reported an interesting case of a 14-year-old girl with vitiligo who was hospitalized for hypotensive shock following COVID-19 infection, initially treated as MIS-C and afterward discovered to be autoimmune thyroiditis and primary adrenal insufficiency. The concomitance of these autoimmune conditions led to the diagnosis of autoimmune polyglandular syndrome 2 (APS2) [28]. The role of COVID-19 in the ethiopathogenesis of APS2 is unclear, but it might be the trigger of the rapid progression of both adrenal insufficiency and hypothyroidism [28].

A third potential mechanism is a selective transient pituitary dysregulation, secondary either to the direct cytotoxic effect of the virus on the hypophysis and an indirect effect of the cytokine storm that would induce NTIS [29,30,31,32].

It must be also taken into account that the COVID-19 pandemic reduced the general population’s access to health services worldwide, and this could have had an impact on the severity of thyroid disease at diagnosis in adults [33], as also suggested by a study conducted in the United Kingdom, where nearly 1/3 of children with trisomy 21 did not receive the recommended annual TSH screening in 2020, in comparison with 3% in 2015 [34].

## 4. Adrenal Glands

COVID-19 infection in children has been associated with adrenal gland involvement. As reported, the infection can cause primary adrenal insufficiency, mediated by the activation of cytokines, such as IL-6 and toll-like receptors (TLR), and secondary failure due to glucocorticoids, one of the main treatments for children affected by MIS-C. A retrospective study conducted in a tertiary pediatric hospital in the U.K. collected data from a population of more than 100 children with MIS-C; prolonged and high-dose steroid therapy is often used in severe MIS-C and it can cause hypothalamic–pituitary–adrenal (HPA) axis suppression [35]. A case of an 11-year-old Turkish boy with adrenal insufficiency during MIS-C is reported in the literature; adrenal function was still impaired at a 4-month follow-up [36]. Flokas et al. [28] reported the case of a 14-year-old girl with vitiligo, with new-onset of autoimmune polyglandular syndrome type 2 (hypothyroidism and primary adrenal insufficiency) shortly after SARS-CoV-2 infection. The authors concluded that it would be prudent to monitor autoimmune conditions triggered by SARS-CoV-2 infection, especially in patients with previous autoimmune diseases [28].

On the other hand, patients who already suffer from primary or secondary adrenal insufficiency have an increased vulnerability to infections, including to COVID-19; SARS-CoV-2 infection could trigger a potential life-threatening condition such as an adrenal crisis due to an increased demand of glucocorticoids, both in adults and children. Close monitoring of these patients is then recommended. In particular, in cases of mild to moderate infection, clinical practice guidelines recommend to double or triple the usual daily hydrocortisone dose; in cases of severe infection or profuse vomiting, medical evaluation is mandatory to initiate parenteral administration of glucocorticoids. Only close monitoring is recommended in asymptomatic patients [37]. A multicentric study, conducted in 12 centers in 8 European countries between January 2020 and December 2021, reported data from 64 patients (13 of those children) with adrenal insufficiency and acute SARS-CoV-2 infection; the clinical outcome in this population appeared good with appropriate glucocorticoid dose adjustments [38].

Even if there is evidence in the literature that the population with adrenal insufficiency is at higher risk of SARS-CoV-2 infection, which can lead to adrenal crisis, data are still conflicting regarding the more severe clinical course of the COVID-19 disease in these patients. Banull et at. [39] conducted a retrospective study, collecting data of 390 pediatric patients, concluding that children with a pre-existing endocrine condition, such as diabetes mellitus, adrenal insufficiency, or hypothyroidism, can have a more severe clinical presentation in case of SARS-CoV-2 infection, with a higher risk of hospitalization and/or intensive care unit admission. Chien et al. [40] reported a case of a 17-year-old boy with a positive test for SARS-CoV-2, fever, and neurological symptoms (altered mental status, seizures), with a history of surgical resection of a craniopharyngioma; a prompt hydrocortisone dose rapidly resolved neurological symptoms. The authors concluded that early recognition and treatment with corticosteroids can prevent long-term consequences, especially in patients with brain tumors or Addison’s disease. An Italian study also reported the case of a 9-year-old child infected with SARS-CoV-2 with a history of suprasellar nongerminomatous germ cell tumor, diabetes insipidus, and hypothalamic-pituitary failure [41].

Also, children may suffer from tertiary adrenal insufficiency due to chronic treatment with corticosteroids for other medical conditions and subsequent iatrogenic immune impairment, with an increased risk of SARS-CoV-2 infection and more severe clinical course [37]. So, endogenous conditions leading to hypercortisolism, such as Cushing syndrome, may also represent a risk factor for SARS-CoV-2 infection and a severe course of the disease. There are no available data in the literature on the association between COVID-19 and hypercortisolism in children. Furthermore, it is important to consider that chronic cortisol excess leads not only to immune impairment but also to comorbidities (hypertension, obesity, and hyperglycemia), which are associated with a higher risk of severe COVID-19 disease. 

To summarize, SARS-CoV-2 infection may be responsible, in children, for both direct and indirect HPA axis alteration due to glucocorticoids treatment in case of MIS-C. On the other hand, children who already have an impairment of adrenal function, insufficiency or hypercortisolism need careful monitoring in case of COVID-19 disease.

## 5. Hypothalamic–Pituitary Involvement

In the literature, there are very limited data about the association between SARS-CoV-2 and the hypothalamic–pituitary axis in children and adolescents.

As with its predecessors, SARS-CoV-2 may enter the central nervous system through the olfactory bulb, reaching hypothalamus and hypophysis [41], which constitute putative targets for this virus due to the expression of ACE-2 receptors and TMPRSS2 on the surfaces of hypothalamic and pituitary cells [42,43,44].

Different injury mechanisms regarding COVID-19 infection and the hypothalamic–pituitary axis have been hypothesized in the literature: direct hypothalamic injury induced by the virus itself, reversible immune-mediated hypophysitis, molecular mimicry between specific virus sequences and ACTH with subsequent cross-reaction, massive cytokine production that reduces ACTH release, decreasing its effect on adrenal tissue [44,45,46,47,48,49,50].

### 5.1. Pituitary Gland Involvement of SARS-CoV-2 Virus

One of the few cases reported in the literature of a potential involvement of hypophysis by SARS-CoV-2 in children is for a 4-year-old girl with a rapid-onset obesity with hypothalamic dysfunction, hypoventilation, and autonomic dysregulation (ROHHAD)-syndrome-like phenotype after COVID-19 infection: she developed electrolyte alterations (hypernatremia and hyperchloremia), hypocorticism and hypothyroidism, central hypoventilation, bulimia, and progressive obesity with metabolic disimpairment. The MRI of the brain showed mild posthypoxic changes, and the condition was resolved with nonsteroidal anti-inflammatory drugs and monthly courses of intravenous immunoglobulin [51].

The second case of the involvement of hypophysis of the virus concerns an 18-year-old previously healthy girl who presented with symptomatic lymphocytic hypophysitis three weeks after COVID-19 (the first ever reported in the literature). She complained of acute onset headache and dizziness for 5 days. This condition was documented with a brain MRI and treated with methylprednisolone 250 mg IV every 6 h on days 1–3; on day 3, they observed symptomatic clinical improvement with a significant decrease in the intensity of the headaches [52].

### 5.2. SARS-CoV-2 Infection in Children with Pre-Existing Hypothalamic–Pituitary Axis Dysfunction

There is no evidence that children affected by hypopituitarism present a higher risk of contracting COVID-19 infection or experiencing a severe disease course, not even patients affected by secondary adrenal insufficiency [40,53].

As evidence of this, R. Gaudino et al. [40] reported the case of a 9-year-old boy with SARS-CoV-2 and a recent diagnosis of suprasellar nongerminomatous germ cell tumor, also suffering from diabetes insipidus and hypothalamic–pituitary failure, who remained asymptomatic for the duration of the infection without requiring any change in his habitual replacement therapy.

What has emerged in these years, however, is that patients affected by hypopituitarism and secondary adrenal insufficiency potentially have a higher risk of undergoing adrenal crisis during the SARS-CoV-2 infection, as with any other infection [13].

### 5.3. Growth Hormone (GH) Release and Precocious Puberty during SARS-CoV-2 Pandemic

During COVID-19, we have also witnessed two opposing phenomena: a reduction in pediatric GH deficiency and an increase in central precocious puberty (CPP) diagnoses [54].

Regarding tests investigating pediatric GH deficiency, Peinkhofer et al. [54] detected a striking reduction (−35%) in tests performed in 2020 compared with 2019, in contrast with the trend of previous years, which consisted of an in increase in referrals for growth issues. Some hypotheses postulated to explain this phenomenon are that both pediatricians and families had less opportunity to detect short stature and delayed growth because of fewer well-child visits and fewer chances to compare their children with classmates; moreover, some appointments were probably cancelled during pandemic due to the tendency to avoid hospitals [54].

Regarding CPP diagnoses, COVID-19 infection in children has been associated with several endocrine disorders.

Many retrospective and cross-sectional studies have reported a significant increase during the pandemic in central CPP diagnoses and in pubertal progression rate in patients already diagnosed with CPP compared with the prepandemic period [54,55,56,57,58,59,60,61,62,63,64,65,66,67,68,69,70,71,72,73,74,75,76,77,78,79,80].

The first retrospective study reporting an increase in CPP diagnoses and a faster rate of pubertal progression during the first lockdown (March–July 2020), compared with the same period in the previous five years, was conducted by Stagi et al. [76]. The first hypothesis was that BMI increase and electronic device overuse were both significantly higher in the pandemic group [76]. In following retrospective Italian studies, no significant differences in BMI increase were found in patients diagnosed with CPP during the pandemic [54,79,80], but Umano et al. reported sleep disturbances as a frequent comorbidity [80].

The same trend was reported all over the world. In a retrospective study by Fu et al., the data from 22 medical institutions in China showed that 4281 female patients were diagnosed with new-onset precocious puberty between February and May 2020, five times more than observed in the same period in 2018 and three times more than observed in the same period in 2019; the authors also reported weight and BMI were significantly higher in the 2020 group [70]. All the studies cited above reported the same increase in CPP diagnoses, especially in women, and a faster rate of pubertal progression, but a correlation with a BMI increase during the pandemic was not always found.

Many authors then tried to explain this new trend. SARS-CoV-2 infection may have direct and indirect effects on puberty. Focusing on direct effects, three main possible mechanisms have been considered, but they need to be investigated further as they are still poorly understood: direct inflammation of the olfactory bulb, which shares a common embryogenic origin with the hypothalamic GnRH neurons; blood–brain barrier disruption; and cytokine storm [81]. The indirect effects of SARS-CoV-2 probably played a more significant role in this outbreak of CPP diagnoses, and they consist of physical and psychosocial changes due to lockdown and social distancing during the pandemic. Among the social effects, stress fear and anxiety linked to social distancing and lack of physical activity, family mourning, and the worldwide burden of pandemic might have played a fundamental role. In fact, these psychological conditions may activate GABA A receptors and consequently the stress pathways responsible for puberty onset [82]. Regarding physical changes, reductions in physical activity, increased BMI, and glucose and insulin metabolism alterations have been widely reported in children and adolescents during the pandemic, and obesity has been associated with the secular trend in puberty anticipation [83]. Also, even hyperinsulinemia not associated with obesity has been correlated with precocious puberty, since insulin resistance may be responsible for an increased bioavailability of sex hormones [84]. Other factors that may be associated with this trend include increased screen time due to online school classes and a reduction in outdoor activities, vitamin D deficiency [85] (which has already been associated with precocious puberty, especially in girls), and sleep disorders [86].

Two recent retrospective studies help us to better understand the correlation between CPP and SARS-CoV-2 infection. The first one, conducted by Goffredo et al., showed that children with precocious puberty presented lower bone age advancement and higher levels of 17-hydroxyprogesterone than those observed prelockdown. These findings strongly suggest the influence of newly emerged environmental factors on pubertal development, although no study has been able to demonstrate a single cause–effect association until now [58].

The second one [68] registered the number of girls with suspected precocious puberty and the relative percentage of rapidly progressive CPP from 2019 to 2022. The authors found a significant increase in both of these diagnoses during 2020 compared with 2019 and a gradual decrease in the same diagnoses in 2021 and 2022, concurrent with the progressive resumption of daily activities. These findings suggest, once again, that radical lifestyle changes and the consequent stress due to the COVID-19 lockdown that children and adolescents underwent might have been crucial in regulating pubertal timing during the pandemic [68].

To summarize, the direct effect of SARS-CoV-2 infection on CPP onset remains unclear, but physical and psychological aspects related to the pandemic may have triggered a GnRH pulsatile secretion, leading to CPP.

## 6. Diabetes

The association between COVID-19 infection and type 1 diabetes (T1D) in children has been extensively studied in the last three years. However, the available data regarding the trend of incidence of newly diagnosed T1D during the COVID-19 pandemic are conflicting [87,88].

Several studies report a clear increase in the incidence of T1D since the beginning of the worldwide spread of the SARS-CoV-2 virus [89,90,91,92,93,94,95,96,97,98,99,100,101,102,103,104,105,106,107,108,109,110,111,112,113]. According to a meta-analysis by Rahmati et al. [114], the global new-onset of childhood T1D rate in 2020 was 32.39 per 100,000 children, clearly higher in comparison with 2019 (19.73 per 100,000 children) [114]. In a cohort study in seven U.S. centers, T1D was reported in 781,419 children and adolescents aged 0–17 years with laboratory-confirmed COVID-19 [115]. However, most of the above-mentioned studies did not find a direct cause–effect link between COVID-19 infection and T1D onset [89,90,91,92,93,94,95,96,97,98,99,100,101,102,103,104,105,106,107,108,109,110,111].

A case–control study conducted on children and adults with and without T1D in Colorado, USA, showed no difference in the prevalence of SARS-CoV-2 antibodies in the two groups during 2020 [116]. Similar results were reported in a study conducted on subjects aged <16 years in Belgium, in whom SARS-CoV-2 serology was tested within the first month from diabetes onset [117]. Conversely, other centers have reported a previous coronavirus infection or precise exposure to the virus in most of their patients newly diagnosed with T1DM [112,113].

In agreement with these studies, the CDC analyzed IQVIA healthcare data from March 2020 to February 2021 and estimated diabetes incidence among patients aged <18 years with proven COVID-19 infection. This incidence was found to be significantly higher than the incidence of T1D in children without COVID-19 (hazard ratio = 2.66, 95% CI = 1.98–3.56) [118,119].

Given the present data, it has been also hypothesized that SARS-CoV-2 might stimulate the autoimmune system, especially for pancreatic autoimmunity, therefore triggering the onset of T1D [113,114].

Other authors did not find any change in the incidence of T1D during the pandemic [119,120,121,122,123,124], but rather a change in the seasonal pattern, as suggested by the data from the Worldwide SWEET Registry [122]. In fact, in 2020, a shift in the seasonality of T1D incidence, with more cases in the summer months, was observed [122]. This may have been due to the hygiene measures and social distancing adopted during the lockdown, which may have further reduced the prevalence of viral infections in the winter/spring season, decreasing the impact of viral triggers for T1D onset in potentially susceptible subjects [122].

Some studies describe a decreased incidence of T1D after the beginning of the pandemic [125,126]. A possible explanation for this finding, in addition to the increased protection measures against COVID-19, which reduced the risk of contracting the most common viral forms, is the increased expression of CD8 + lymphocytes in T1D, which may protect these patients against infections [125,127].

Data about the prevalence of diabetic ketoacidosis (DKA) are much more concordant in the literature. Several cohort studies [91,93,98,102,106,123,126,127,128,129,130,131,132,133,134,135,136,137,138,139,140,141,142,143,144,145,146,147,148,149,150,151,152,153,154,155,156,157,158,159,160,161,162,163] and two meta-analyses [114,164] report increases in DKA frequency and DKA severity at diagnosis among newly diagnosed T1D patients. A multicenter study, involving 13 national diabetes registries (Australia, Austria, Czechia, Denmark, Germany, Italy, Luxembourg, New Zealand, Norway, Slovenia, Sweden, the USA (Colorado), and Wales) compared the observed DKA prevalence in children and adolescents in 2020 and 2021 to predictions based on trends over the prepandemic years 2006–2019. The pre-existing increase in the prevalence of DKA at diagnosis of T1D in children and adolescents from 2006 to 2019 was exacerbated in 2020 and 2021 [165].

To better study the association between SARS-CoV-2 and DKA prevalence at T1D onset, Kamrath et al. [166] calculated the relative risk (RR) of DKA at diagnosis of T1D during the year 2020 to assess whether it was associated with the regional incidence of COVID-19 cases and deaths. The applied multivariable mixed-effects log-binomial model revealed a significant association between the regional weekly incidences of COVID-19 cases and COVID-19-related deaths and the corresponding rates of ketoacidosis at diagnosis of T1D during the first half of the year 2020 [166].

Only a few centers did not observe a significant difference between the prepandemic and postpandemic era [94,105,108,167]. This finding has been attributed to increased parental supervision during the pandemic, which might have prevented severe disease decompensation in some cases [167].

Some authors have attributed the higher frequency of DKA to delayed diagnosis and initiation of insulin replacement therapy [168,169]. It has been suggested that elevated HbA1c at diagnosis of T1D, beyond an increase in the prevalence of DKA at onset, may be the consequence of a delay in diagnosis and insulin treatment initiation secondary to the pandemic difficulties [136]. Moreover, a retrospective cohort study on seven U.S. centers observed the highest frequency of DKA among T1D diabetes during COVID-19 surges in non-Hispanic Black patients, bringing attention to the problem of reduced access to health care during the pandemic by patients who are victims of health inequities [115].

To better understand the association between SARS-CoV-2 infection and DKA, a retrospective study compared insulin-mediated tissue glucose disposal (TGD) during standardized therapy for DKA in all children with pre-existing T1D with or without COVID-19. The median TGD was 46% lower among patients with COVID-19 infection in comparison with those without the infection. These results suggest that SARS-CoV-2 is associated with greater insulin resistance in DKA among patients affected by T1D, leading to the hypothesis that COVID-19 causes a metabolic impairment beyond factors that typically contribute to pediatric DKA [170].

To summarize, although the literature is discordant, the evidence of a potential role of SARS-CoV-2 in T1D onset and DKA frequency in newly diagnosed and known T1D diabetes is quite consistent, which deserves further investigation to understand the underlying responsible mechanisms. It is well known that some viral infections, especially rubella, Coxsackie, mumps, enterovirus, and the Epstein–Barr virus, represent important environmental risk factors for the development of T1D [171,172]. Since the start of the SARS-CoV-2 pandemic, attention has been paid on this novel virus as a further infective cause in the development and progression of T1D [173]. In fact, beyond the organizational problems related to the lockdown and the decreased access to health services during the pandemic [174], some biological mechanisms have been hypothesized to contribute to its pathogenesis [175].

Two main mechanisms are known to be responsible for the damage caused to pancreatic cells by different viruses. The first one is the direct cytolysis of virally infected cells; the second one is an autoimmune reaction [176]. The role of COVID-19 still needs to be clarified; however, pancreatic cell damage seems to occur, especially in patients with severe SARS-CoV-2 infection [174,177,178,179,180]. Nevertheless, the presence of the virus has been demonstrated only in the pancreas of deceased patients [181,182]. Rubino et al. [177] suggested that SARS-CoV-2 may bind to its cellular entry ACE-2 receptors, even in the pancreatic beta cells, where these cells are abundant, leading to pancreatic beta cell destruction. Another theory that could explain the association between SARS-CoV-2 and T1D development and the worse presentation at onset is virus-induced aberrant immune response, which may attack the pancreatic islet cells, mimicking the pathogenesis of insulin-dependent diabetes mellitus [183]. Indeed, cytokine storm, commonly found in COVID-19, may impair β-cell insulin secretion and glucose control [184]. However, the latter hypothesis was tested by Rewers et al. [185] by measuring autoantibodies to insulin, glutamic acid decarboxylase, islet antigen 2, and zinc transporter 8 autoantibodies in children both with and without previous COVID-19 infection, defined by the presence of antibodies to both SARS-CoV-2 receptor binding domain and nucleocapsid proteins. No association of SARS-CoV-2 infection with autoimmunity related to the development of T1D was observed in 50.000 youths from different populations in Colorado and Bavaria [185].

Another recently advanced hypothesis is that SARS-CoV-2 may cause T1D through the interferon-α-activated latent ribonuclease (RNAseL) signaling pathway [186]. Excessive RNaseL activity may lead to the degradation of both pathogen and host RNA, leading to cellular damage. This activity is regulated by phosphodiesterases such as PDE12. PDE12 expression has been shown to be decreased in individuals with recently diagnosed T1D. This seems to have a protective effect against viral infections, upregulating RNaseL activity; however, a potential side effect is a trigger of beta-cell damage [186].

MIS-C, as a severe complication of SARS-CoV-2 in children, may also be responsible for the pathogenesis of insulin resistance (IR). An exploratory study analyzed the glycemic patterns of 30 normal-weight children affected by MIS-C, showing a high prevalence of IR and glycemic fluctuation [187]. Data on the persistence of IR have been also described [188]. These evidence suggest that regular glucose monitoring of both fasting and postprandial glucose levels should be performed in patients with MIS-C.

## 7. Limits in the Scope of the Study

As mentioned in the Materials and Methods, our narrative review has some limits due to the selection method of the articles. Language was restricted to English only to ensure authors’ complete understanding of the articles analyzed. Furthermore, to standardize the search, it was decided to use PubMed and Embase as databases. We are aware that this selection may have resulted in the exclusion of some documents reporting different results from those shown. The above-mentioned factors, together with the relatively small number of available articles in the literature, represent a limit in the scope of our study.

## 8. Conclusions

Although little is known regarding the involvement of the endocrine system during COVID-19 in children; the current evidence in adults and epidemiological studies on the pediatric population suggest the presence of a causal association between the virus and endocrinopathies.

The available data support the hypothesis that SARS-CoV-2 is responsible for transitory thyroid dysfunction. Thus, it would be a good practice to assess thyroid function markers in children affected with COVID-19 infection, especially if with moderate or severe manifestations and MIS-C.

Furthermore, the virus might affect the HPA axis in both direct and indirect ways due to glucocorticoids treatment in case of MIS-C. Also, it is recommended to pay special attention to children with known impairment of adrenal function who experience SARS-CoV-2 infection, because they may present with a more severe clinical picture than healthy children.

The hypothalamic–pituitary axis may also be affected in different ways: a case of ROHHAD-syndrome-like presentation and a case of hypohysitis after COVID-19 infection have been reported in children. Moreover, an increase in CPP incidence has been observed since the advent of the COVID-19 pandemic. The direct effect of the virus in the pathogenesis of CPP is still unclear; however, it has been suggested that its related physical and psychological aspects may act as triggers of GnRH pulsatile secretion.

Regarding T1D, despite some discordant evidence, many cohort studies support the role of SARS-CoV-2 in T1D onset and DKA frequency in newly diagnosed and known T1D diabetes. It has been suggested that this association is due to the combined direct effects of the virus on pancreatic cells, the indirect effects of the virus through its consequent inflammatory response, and delayed diagnoses secondary to the healthcare difficulties encountered worldwide. An association between IR and MIS-C should be also considered.

In conclusion, although data regarding the relationship between COVID-19 and endocrinological involvement in pediatric patients are limited, they suggest the correlation between SARS-CoV-2 infection and impairment in different organs in this population, as more extensively reported in adults Therefore, further studies are needed to clarify the role of SARS-CoV-2 infection in the development of different endocrinopathies in children, both during acute infection and as a result of its long-term complications. Moreover, health institutions should be aware of this likely association and highlight endocrine involvement in this population of interest. This information is of primary importance to guide the management of patients with previous or current COVID-19.

## Figures and Tables

**Table 1 jcm-12-05248-t001:** List of keywords used in our literature search categorized by endocrine system or organ and respective type and number of articles found.

Endocrine System or Organ	Keywords	Number of Suitable Articles (Total Number)	Type of Article
Thyroid	“thyroid” OR “hyperthyroidism” OR “hypothyroidism” OR “thyroiditis” AND “coronavirus disease 2019” OR “COVID-19” OR “SARS-CoV-2” OR “MIS-C” OR “multisystem inflammatory syndrome” AND “children” OR “adolescents”	12 (33)	1 cross-sectional study 3 cohort studies 1 case control study 1 retrospective case note review 1 cross-sectional chart review 2 narrative reviews 3 case reports
Adrenal glands	“adrenal insufficiency” OR “hypercortisolism” AND “coronavirus disease 2019” OR “COVID-19” OR “SARS-CoV-2” OR “MIS-C” OR “multisystem inflammatory syndrome” AND “children” OR “adolescents”	10 (10)	2 narrative reviews 2 single-center or multicenter cohort studies 1 cross-sectional study 1 clinical practice guideline 4 case reports
Hypothalamus-pituitary system	“hypothalamus” OR “pituitary” OR “Hypopituitarism” OR “hypophysitis” OR “growth hormone” OR “central precocious puberty” AND “coronavirus disease 2019” OR “COVID-19” OR “SARS-CoV-2” OR “MIS-C” OR “multisystem inflammatory syndrome” AND “children” OR “adolescents”	34 (96)	4 cross-sectional studies 1 case control study 17 retrospective studies 9 reviews 3 case reports
Gonads	“puberty” OR “central precocious puberty” AND “coronavirus disease 2019” OR “COVID-19” OR “SARS-CoV-2” OR “MIS-C” OR “multisystem inflammatory syndrome” AND “children” OR “adolescents”	32 (32)	5 narrative reviews 1 cohort study 6 cross-sectional studies 20 case-control studies
Pancreas	“type 1 diabetes” OR “juvenile onset diabetes” OR “insulin-dependent diabetes” OR “T1D” AND “coronavirus disease 2019” OR “COVID-19” OR “SARS-CoV-2” OR “MIS-C” OR “multisystem inflammatory syndrome” AND “children” OR “adolescents”	102 (253)	2 meta-analyses 6 narrative reviews 72 cohort studies 9 cross-sectional studies 2 case-control studies 11 case reports or case series

## Data Availability

Not applicable.

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
