# Peer review of "Endocrinological Involvement in Children and Adolescents Affected by COVID-19: A Narrative Review"

_jcm, 2023, doi:10.3390/jcm12165248_

Round 1

Reviewer 1 Report

Article that clearly exposes the impact of the COVID-19 pandemic on pediatric endocrinology by offering an interesting update. The review seems broad and well divided into various paragraphs that deal in detail with the various endocrinological systems.

Some minor modifications:

1)     In table 1 I can not read the number of the selected articles (as suggested in the legend)

2)     In table 1 as well I would homologate the second and the third part of the key words, for examples sometimes there is MISC and sometimes there is not.

English is correct and exposition clear.

Author Response

Reviewer 1

  • In table 1 I cannot read the number of the selected articles (as suggested in the legend)

R: We’ve added the number and type of found articles in table 1 as required.

  • In table 1 as well I would homologate the second and the third part of the key words, for examples sometimes there is MISC and sometimes there is not.

R: We’ve homologated the second and third part of the key words as required.

Reviewer 2 Report

The work is very interesting, authors summarized the published report on some endocrine glands 

Some comments are listed here to improve the manuscript   

Major comments

1.     COVID 19 infections were not discussed many endocrine organs, so the content didn’t reflect the title . As the endocrine system doesn’t include thyroid, adrenal, pancreas and pituitary but it includes other important glands such as parathyroid gland, pineal gland, gonads, thymus, and other tissues that can act as endocrine organs or tissues such as fat tissue, kidney, intestine, skin brain and others ….Therefore authors have to add separate sections for those endocrine organs and tissue or one comprehensive section on the missing organs. Or authors rescope and concentrate to revise the work to be written on those glands in specific way only with revise the title and the abstract 

2.    Lines 113-121: There is no clear link between high IL-6 and thyroid or with low fT3 levels in those patients which didn’t give meaning for this conclusion. 

3.    Lines 123-127 have to be rewritten 

4.    Lines 136-137: Were they COVID 19’ patients ? 

5.    Line 161L: please add the full name of APS2 and all abbreviation at first time of their mention

Line 171: you can cite this recent study ref to promote your conclusion DOI:https://doi.org/10.1016/S2213-8587(23)00094-3

6.    Line 182: correct report to reported and please correct other verbs 

7.     Lines 221: This reference is  found to b esuitable to enrich this section as authors said “The only available data in literature on hypercortisolism involve adults [41]” 

8.     230 – 231: In literature there is very limited data about the association between SARS-CoV-2 and hypothalamic-pituitary axis. 

I searched simply in PUBmed and found many references such as this list 

Implications of testicular ACE2 and the renin–angiotensin system for SARS-CoV-2 on testis function

R. Clayton Edenfield, Charles A. Easley, IV

Nat Rev Urol. 2022; 19(2): 116–127. Published online 2021 Nov 26. doi: 10.1038/s41585-021-00542-5

SARS-CoV-2 Enters Human Leydig Cells and Affects Testosterone Production In Vitro

Lu Li, Chantal M. Sottas, Hsu-Yu Chen, Yuchang Li, Haoyi Cui, Jason S. Villano, Joseph L. Mankowski, Paula M. Cannon, Vassilios Papadopoulos

Cells. 2023 Apr; 12(8): 1198. Published online 2023 Apr 20. doi: 10.3390/cells12081198

Analysis of SARS-CoV-2 synonymous codon usage evolution throughout the COVID-19 pandemic

Ezequiel G. Mogro, Daniela Bottero, Mauricio J. Lozano

Virology. 2022 Mar; 568: 56–71. Published online 2022 Feb 2. doi: 10.1016/j.virol.2022.01.011

Viral pathogenesis of SARS-CoV-2 infection and male reproductive health

Shubhadeep Roychoudhury, Anandan Das, Niraj Kumar Jha, Kavindra Kumar Kesari, Shatabhisha Roychoudhury, Saurabh Kumar Jha, Raghavender Kosgi, Arun Paul Choudhury, Norbert Lukac, Nithar Ranjan Madhu, Dhruv Kumar, Petr Slama

Open Biol. 2021 Jan; 11(1): 200347. Published online 2021 Jan 20. doi: 10.1098/rsob.200347

.

SARS-CoV-2 (COVID-19) as a possible risk factor for neurodevelopmental disorders

Harikesh Dubey, Ravindra K. Sharma, Suraj Krishnan, Rebecca Knickmeyer

Front Neurosci. 2022; 16: 1021721. Published online 2022 Dec 16. doi: 10.3389/fnins.2022.1021721

Neurological pathophysiology of SARSCoV2 and pandemic potential RNA viruses: a comparative analysis

Nikhil Chakravarty, Thrisha Senthilnathan, Sophia Paiola, Priya Gyani, Sebastian Castillo Cario, Estrella Urena, Akash Jeysankar, Prakash Jeysankar, Joseph Ignatius Irudayam, Sumathi Natesan Subramanian, Helen Lavretsky, Shantanu Joshi, Gustavo Garcia, Jr., Arunachalam Ramaiah, Vaithilingaraja Arumugaswami

FEBS Lett. 2021 Dec; 595(23): 2854–2871. Published online 2021 Nov 22. doi: 10.1002/1873-3468.14227

SARS-CoV-2 Psychiatric Sequelae: A Review of Neuroendocrine Mechanisms and Therapeutic Strategies

Mary G Hornick, Margaret E Olson, Arun L Jadhav

Int J Neuropsychopharmacol. 2022 Jan; 25(1): 1–12. Published online 2021 Oct 14. doi: 10.1093/ijnp/pyab069

SARS-CoV-2 Infection in Pregnant Women: Neuroimmune-Endocrine Changes at the Maternal-Fetal Interface

Marcelo Gomes Granja, Amanda Candida da Rocha Oliveira, Camila Saggioro de Figueiredo, Alex Portes Gomes, Erica Camila Ferreira, Elizabeth Giestal-de-Araujo, Hugo Caire de Castro-Faria-Neto. Neuroimmunomodulation. 2021 May; 28(1): 1–21. Published online 2021 Apr 28. doi: 10.1159/000515556

Non-Productive Infection of Glial Cells with SARS-CoV-2 in Hamster Organotypic Cerebellar Slice Cultures

Lise Lamoureux, Babu Sajesh, Jessy A. Slota, Sarah J. Medina, Matthew Mayor, Kathy L. Frost, Bryce Warner, Kathy Manguiat, Heidi Wood, Darwyn Kobasa, Stephanie A. Booth

Viruses. 2022 Jun; 14(6): 1218. Published online 2022 Jun 3. doi: 10.3390/v14061218

9.     Lines 294-301: authors didn’t show link between covid 19 infection and high CPP or BMR, as these diagnoses could be attributed to other factors such as big and repeated meals, or genetics factor, no more exercises/ school activities or vaccination, which may be attributed to the behaviors during the pandemic. Therefore, I cannot agree with this conclusion unless authors explain the underlying mechanism or direct causes between covid 19 infection and high CPP and BMR. Such as acompaination between GH, cortisol, leptin hormones, insulin resistance and /or gonads hormones in those COVID 19 patients in comparison with health children.  

10.    Lines: 303-336: authors wrote some possible explanation which is acceptable, but they have to separate between the effect of kids behaviors during the pandemic and the effect of COVID virus itself. As lookdown may be happened again because of another pandemic and leads to the same effects, so the COVID 19 virus itself will be not responsible for these diagnoses unless it is proven with lab analysis. So there is significant difference between social effects and medical effects  

11.    Lines 320-322: no link between endocrine disrupting chemicals (EDC) and covid infection as most people used plasticizers during pandemic and non- pandemic periods 

12.    Diabetes section is well written and attributed to physiological and patho endocrinological causes 

13.    Line: 477 it needs editing 

It needs minor editing 

Author Response

Reviewer 2

  • COVID 19 infections were not discussed many endocrine organs, so the content didn’t reflect the title. As the endocrine system doesn’t include thyroid, adrenal, pancreas and pituitary but it includes other important glands such as parathyroid gland, pineal gland, gonads, thymus, and other tissues that can act as endocrine organs or tissues such as fat tissue, kidney, intestine, skin brain and others …. Therefore authors have to add separate sections for those endocrine organs and tissue or one comprehensive section on the missing organs. Or authors rescope and concentrate to revise the work to be written on those glands in specific way only with revise the title and the abstract.

R: thank for your observation, we’ve modified the title and rephrased the aim of the review as required.

  • Lines 113-121: There is no clear link between high IL-6 and thyroid or with low fT3 levels in those patients which didn’t give meaning for this conclusion.

R: We resised the text to clarify

  • Lines 123-127 have to be rewritten

R; We’ve rewritten the sentence as required

  • Lines 136-137: Were they COVID 19’ patients?

R: They were all MIS-C patients, as we have now specified

  • Line 161L: please add the full name of APS2 and all abbreviation at first time of their mention

R: Done

  • Line 171: you can cite this recent study ref to promote your conclusion DOI:https://doi.org/10.1016/S2213-8587(23)00094-3

R: we added te reference [33]

  • Line 182: correct report to reported and please correct other verbs

R: Done

  • Lines 221: This reference is found to be suitable to enrich this section as authors said “The only available data in literature on hypercortisolism involve adults [41]”

We’ve removed this reference, since the aim of our review was to focus on pediatric population.

  • 230 – 231: In literature there is very limited data about the association between SARS-CoV-2 and hypothalamic-pituitary axis.

I searched simply in PUBmed and found many references such as this list:

  • Implications of testicular ACE2 and the renin–angiotensin system for SARS-CoV-2 on testis function. R. Clayton Edenfield, Charles A. Easley, IV Nat Rev Urol. 2022; 19(2): 116–127. Published online 2021 Nov 26. doi: 10.1038/s41585-021-00542-5

  • SARS-CoV-2 Enters Human Leydig Cells and Affects Testosterone Production In Vitro. Lu Li, Chantal M. Sottas, Hsu-Yu Chen, Yuchang Li, Haoyi Cui, Jason S. Villano, Joseph L. Mankowski, Paula M. Cannon, Vassilios Papadopoulos. Cells. 2023 Apr; 12(8): 1198. Published online 2023 Apr 20. doi: 10.3390/cells12081198
  • Analysis of SARS-CoV-2 synonymous codon usage evolution throughout the COVID-19 pandemic. Ezequiel G. Mogro, Daniela Bottero, Mauricio J. Lozano. Virology. 2022 Mar; 568: 56–71. Published online 2022 Feb 2. doi: 10.1016/j.virol.2022.01.011
  • Viral pathogenesis of SARS-CoV-2 infection and male reproductive health. Shubhadeep Roychoudhury, Anandan Das, Niraj Kumar Jha, Kavindra Kumar Kesari, Shatabhisha Roychoudhury, Saurabh Kumar Jha, Raghavender Kosgi, Arun Paul Choudhury, Norbert Lukac, Nithar Ranjan Madhu, Dhruv Kumar, Petr Slama. Open Biol. 2021 Jan; 11(1): 200347. Published online 2021 Jan 20. doi: 10.1098/rsob.200347
  • SARS-CoV-2 (COVID-19) as a possible risk factor for neurodevelopmental disorders. Harikesh Dubey, Ravindra K. Sharma, Suraj Krishnan, Rebecca Knickmeyer. Front Neurosci. 2022; 16: 1021721. Published online 2022 Dec 16. doi: 10.3389/fnins.2022.1021721
  • Neurological pathophysiology of SARS‐CoV‐2 and pandemic potential RNA viruses: a comparative analysis. Nikhil Chakravarty, Thrisha Senthilnathan, Sophia Paiola, Priya Gyani, Sebastian Castillo Cario, Estrella Urena, Akash Jeysankar, Prakash Jeysankar, Joseph Ignatius Irudayam, Sumathi Natesan Subramanian, Helen Lavretsky, Shantanu Joshi, Gustavo Garcia, Jr., Arunachalam Ramaiah, Vaithilingaraja Arumugaswami. FEBS Lett. 2021 Dec; 595(23): 2854–2871. Published online 2021 Nov 22. doi: 10.1002/1873-3468.14227
  • SARS-CoV-2 Psychiatric Sequelae: A Review of Neuroendocrine Mechanisms and Therapeutic Strategies. Mary G Hornick, Margaret E Olson, Arun L Jadhav. Int J Neuropsychopharmacol. 2022 Jan; 25(1): 1–12. Published online 2021 Oct 14. doi: 10.1093/ijnp/pyab069
  • SARS-CoV-2 Infection in Pregnant Women: Neuroimmune-Endocrine Changes at the Maternal-Fetal Interface. Marcelo Gomes Granja, Amanda Candida da Rocha Oliveira, Camila Saggioro de Figueiredo, Alex Portes Gomes, Erica Camila Ferreira, Elizabeth Giestal-de-Araujo, Hugo Caire de Castro-Faria-Neto. 2021 May; 28(1): 1–21. Published online 2021 Apr 28. doi: 10.1159/000515556
  • Non-Productive Infection of Glial Cells with SARS-CoV-2 in Hamster Organotypic Cerebellar Slice Cultures. Lise Lamoureux, Babu Sajesh, Jessy A. Slota, Sarah J. Medina, Matthew Mayor, Kathy L. Frost, Bryce Warner, Kathy Manguiat, Heidi Wood, Darwyn Kobasa, Stephanie A. Booth. Viruses. 2022 Jun; 14(6): 1218. Published online 2022 Jun 3. doi: 10.3390/v14061218

R: Our sentence is referred to pediatric age (we specified in the text). The aforementioned articles are all focused on adults or do not meet our search criteria regarding endocrine involvement in children and adolescents; therefore, we preferred not include these manuscripts in our review. 

  • Lines 294-301: authors didn’t show link between covid 19 infection and high CPP or BMR, as these diagnoses could be attributed to other factors such as big and repeated meals, or genetics factor, no more exercises/ school activities or vaccination, which may be attributed to the behaviors during the pandemic. Therefore, I cannot agree with this conclusion unless authors explain the underlying mechanism or direct causes between covid 19 infection and high CPP and BMR. Such as acompaination between GH, cortisol, leptin hormones, insulin resistance and /or gonads hormones in those COVID 19 patients in comparison with health children.

R: thank you for your observation. We included some possible underlying mechanisms of the association between SARS-CoV-2 and endocrine involvement in lines 310-314. As specified in the paragraph, they are just hypotheses and they are not deeply reported, since these mechanisms still need to be understood. Indirect effects linked to pandemic logistic are mentioned as well.

  • Lines: 303-336: authors wrote some possible explanation which is acceptable, but they have to separate between the effect of kids behaviors during the pandemic and the effect of COVID virus itself. As lookdown may be happened again because of another pandemic and leads to the same effects, so the COVID 19 virus itself will be not responsible for these diagnoses unless it is proven with lab analysis. So there is significant difference between social effects and medical effects.

R: Regarding indirect effects, lines 316-328 have been rephrased in order to distinguish between social and medical effects, as suggested.  

  • Lines 320-322: no link between endocrine disrupting chemicals (EDC) and covid infection as most people used plasticizers during pandemic and non- pandemic periods

R: We’ve removed the mention of endocrine disrupting in the list of possibly contributing factors as suggested.

  • Diabetes section is well written and attributed to physiological and patho endocrinological causes

R; Thank you for your positive comment

  • Line: 477 it needs editing

R: Done

Reviewer 3 Report

In general, the document has a good structure and development of the topic, however, below are some suggestions for improvement of the manuscript.

In the title: It should be improved by indicating the type of review that was performed (systematic, narrative, review and meta-analysis). In addition, it is suggested to indicate a plausible connection between the variables of interest, for example: correlation, alteration, change, disruption, so that the title denotes the important point of this relationship. 

In the keywords, it would be worthwhile to indicate: children, adolescents. 

Throughout the document, care should be taken to use only one type of language, in this case US English or UK English, but not a mixture of both (pediatric, paediatric). 

Perhaps the presence of a table that includes the subtitles, the most important documents with important relationships between the variables, would help a lot to denote the importance of the document, in order to facilitate its reading and highlight what is really important.  

The conclusion section has a touch of discussion, is somewhat vague and not very conclusive, with all the information gathered the authors could indicate that there is, although limited and scarce, information that suggests a correlation, relationship, or interaction between COVID-19 and alterations in one, two, or more of the organs, systems, or endocrine changes analyzed, which in itself would suggest extending the search for literature, or directing efforts for health institutions to contemplate and highlight the endocrine changes analyzed, two, or more of the organs, systems, or endocrine changes analyzed, which in itself would suggest extending the literature search, or directing efforts for health institutions to contemplate and highlight these endocrine changes in this population of interest.  

A small section or subtitle should be placed indicating "limits in the scope of the study", where it is emphasized that the manuscript only took certain language, from certain databases, (something is mentioned in the part of materials and methods) giving a small explanation of why that decision is taken, since there are databases that are recommended for reviews of any kind (sciencedirect, elsevier, scopus, google academic, scielo), and that most of the advances in the subject of COVID-19 and SARS-COV2 are found in other languages such as Portuguese, Spanish, French, etc.  This is undoubtedly a limitation, and the inclusion of some documents with these characteristics, undoubtedly, would yield results that could differ from those shown. 

On the other hand, a little more detail and attention should be paid to the part of materials and methods, so that other authors can replicate the study or can verify in some way what was done in this manuscript; it should be indicated if the articles were open access, or collected by subscription, the criteria for selection, inclusion, exclusion and elimination, although it is briefly indicated that it is a narrative review, certainly including this information will give a formality and adequate structure to the manuscript, scientifically it is what should be done.

The above is intended to provide a better structure and approach to the arduous work carried out by the authors. 

Throughout the document, care should be taken to use only one type of language, in this case US English or UK English, but not a mixture of both (pediatric, paediatric). 

Author Response

Reviewer 3

  • In the title: It should be improved by indicating the type of review that was performed (systematic, narrative, review and meta-analysis). In addition, it is suggested to indicate a plausible connection between the variables of interest, for example: correlation, alteration, change, disruption, so that the title denotes the important point of this relationship.

R: We’ve modified the title as suggested

  • In the keywords, it would be worthwhile to indicate: children, adolescents.

R: We’ve added “children” and “adolescents” as keywords

  • Throughout the document, care should be taken to use only one type of language, in this case US English or UK English, but not a mixture of both (pediatric, paediatric).

R: Done

  • Perhaps the presence of a table that includes the subtitles, the most important documents with important relationships between the variables, would help a lot to denote the importance of the document, in order to facilitate its reading and highlight what is really important.

R: We’ve enriched table 1 with number and type of selected articles as suggested

  • The conclusion section has a touch of discussion, is somewhat vague and not very conclusive, with all the information gathered the authors could indicate that there is, although limited and scarce, information that suggests a correlation, relationship, or interaction between COVID-19 and alterations in one, two, or more of the organs, systems, or endocrine changes analyzed, which in itself would suggest extending the search for literature, or directing efforts for health institutions to contemplate and highlight the endocrine changes analyzed, two, or more of the organs, systems, or endocrine changes analyzed, which in itself would suggest extending the literature search, or directing efforts for health institutions to contemplate and highlight these endocrine changes in this population of interest.

R: The “Conclusions” section has been enriched as suggested.

  • A small section or subtitle should be placed indicating "limits in the scope of the study", where it is emphasized that the manuscript only took certain language, from certain databases, (something is mentioned in the part of materials and methods) giving a small explanation of why that decision is taken, since there are databases that are recommended for reviews of any kind (sciencedirect, elsevier, scopus, google academic, scielo), and that most of the advances in the subject of COVID-19 and SARS-COV2 are found in other languages such as Portuguese, Spanish, French, etc. This is undoubtedly a limitation, and the inclusion of some documents with these characteristics, undoubtedly, would yield results that could differ from those shown.

We’ve added the “limits in the scope of the study” section as recommended (section 7).

  • On the other hand, a little more detail and attention should be paid to the part of materials and methods, so that other authors can replicate the study or can verify in some way what was done in this manuscript; it should be indicated if the articles were open access, or collected by subscription, the criteria for selection, inclusion, exclusion and elimination, although it is briefly indicated that it is a narrative review, certainly including this information will give a formality and adequate structure to the manuscript, scientifically it is what should be done.

We’ve enriched the “Materials and Methods” part as suggested.

Round 2

Reviewer 2 Report

na

na

Reviewer 3 Report

The suggestions were taken into account, and the manuscript had a remarkable improvement, now it can be considered for publication.